# Association of COVID-19-Associated Pulmonary Aspergillosis with Cytomegalovirus Replication: A Case–Control Study

**DOI:** 10.3390/jof8020161

**Published:** 2022-02-06

**Authors:** Jorge Calderón-Parra, Victor Moreno-Torres, Patricia Mills-Sanchez, Sandra Tejado-Bravo, Isabel Romero-Sánchez, Bárbara Balandin-Moreno, Marina Calvo-Salvador, Francisca Portero-Azorín, Sarela García-Masedo, Elena Muñez-Rubio, Antonio Ramos-Martinez, Ana Fernández-Cruz

**Affiliations:** 1Infectious Diseases Unit, Service of Internal Medicine, Hospital Universitario Puerta de Hierro, 28222 Majadahonda, Spain; victor.moreno.torres.1988@gmail.com (V.M.-T.); pamillssa@gmail.com (P.M.-S.); elmuru@gmail.com (E.M.-R.); aramos220@gmail.com (A.R.-M.); anafcruz999@gmail.com (A.F.-C.); 2Research Institute Puerta de Hierro-Segovia de Aranda (IDIPHSA), 28222 Madrid, Spain; 3Intensive Care Unit, Hospital Universitario Puerta de Hierro, 28222 Majadahonda, Spain; sandratejado@gmail.com (S.T.-B.); balandinmoreno@gmail.com (B.B.-M.); 4Microbiology Service, Hospital Universitario Puerta de Hierro, 28222 Majadahonda, Spain; msromero@salud.madrid.org; 5Pharmacy Service, Hospital Universitario Puerta de Hierro, 28222 Majadahonda, Spain; mcalvos@salud.madrid.org (M.C.-S.); fporteroa@gmail.com (F.P.-A.); sarela.garcia-masedo@salud.madrid.org (S.G.-M.)

**Keywords:** COVID-associated pulmonary aspergillosis, CAPA, prevalence, risk factors, cytomegalovirus infection

## Abstract

**Introduction:** Cytomegalovirus (CMV) infection is a well-known factor associated with invasive aspergillosis in immunocompromised hosts. However, its association with COVID-19-associated pulmonary aspergillosis (CAPA) has not been described. We aimed to examine the possible link between CMV replication and CAPA occurrence. **Methods:** A single-center, retrospective case–control study was conducted. A case was defined as a patient diagnosed with CAPA according to 2020 ECMM/ISHAM consensus criteria. Two controls were selected for each case among critically ill COVID-19 patients. **Results:** In total, 24 CAPA cases were included, comprising 14 possible CAPA and 10 probable CAPA. Additionally, 48 matched controls were selected. CMV replication was detected more frequently in CAPA than in controls (75.0% vs. 35.4%, *p* = 0.002). Probable CMV end-organ disease was more prevalent in CAPA (20.8% vs. 4.2%, *p* = 0.037). After adjusting for possible confounding factors, CMV replication persisted strongly associated with CAPA (OR 8.28 95% CI 1.90–36.13, *p* = 0.005). Among 11 CAPA cases with CMV PCR available prior to CAPA, in 9 (81.8%) cases, CMV replication was observed prior to CAPA diagnosis. **Conclusions:** Among critically ill COVID-19 patients, CMV replication was associated with CAPA and could potentially be considered a harbinger of CAPA. Further studies are needed to confirm this association.

## 1. Introduction

COVID-19-associated pulmonary aspergillosis (CAPA) has recently been recognized as a major complication of critically ill COVID-19 patients [1]. It is estimated that 10–20% of ICU admitted COVID-19 patients eventually develop CAPA [2,3], which is associated with a high mortality rate [3]. COVID-19-mediated pulmonary injury [4], impairment in cellular immune response (“immunoparalysis”) [5], and immunosuppressive drugs used to treat COVID-19 [6] are considered to intervene in CAPA pathogenesis. In contrast with influenza-associated pulmonary aspergillosis, in which angioinvasion would be facilitated by extensive respiratory mucosal damage, dysregulation in T-cell response and antigen presentation would be the main mechanisms that facilitate *Aspergillus* infection in COVID-19 patients [7,8].

On the other hand, cytomegalovirus (CMV) infection and/or disease is recognized as a risk factor for fungal infection in highly immunocompromised hosts, such as solid organ transplantation (SOT) [9] and stem cell transplantation recipients [10,11]. Only recently, in a single-center, retrospective study [12], this association has been described in critically ill patients, including patients with influenza. Although a causal role is still to be demonstrated, several factors could influence the higher risk of fungal infection in patients with CMV infection, including T-cell response dysregulation induced by CMV replication itself [13,14].

Some studies suggest CMV replication is frequent among critically ill COVID-19 patients [15] and, theoretically, could induce further T-cell dysfunction in critically ill COVID-19 patients and increase the risk for CAPA. However, to our knowledge, the association between CMV replication and invasive aspergillosis has not been analyzed in COVID-19 patients to date.

Therefore, our main objective was to examine the possible link between CMV replication and CAPA occurrence. We also intended to describe CAPA characteristics, risk factors, and mortality among our patients.

## 2. Materials and Methods

We performed a single-center, retrospective, case–control study. Our hospital is a 613-bed tertiary teaching hospital in Madrid, with a catchment area of 550.000 inhabitants, with 22 ICU beds, which increased to 64 ICU beds during the first waves of the COVID-19 pandemic.

Cases were identified from a prospective cohort that includes all patients diagnosed with invasive fungal infection at our center since January 2017. A case was defined as a patient diagnosed with CAPA (see definition below) between March 2020 and August 2021. We excluded patients with no determination of plasma CMV reverse-transcriptase–polymerase chain reaction (RT–PCR) available during the CAPA admission.

Controls were selected from a prospective cohort including all ICU admitted patients with COVID-19 and severe acute respiratory distress syndrome (ARDS). Controls were matched by admission date to cases, and they were included only if they had at least one plasma CMV RT–PCR determination available during the ARDS admission. Two controls were selected for each case.

Data were collected from electronic medical records and managed using REDCap electronic capture tools [16], with a license provided to Puerta de Hierro-Segovia de Arana Research Institute (IDISPHSA for its Spanish abbreviation) [17]. Data collected using the REDCap platform were anonymized and included demographics, comorbidities, microbiological data, and outcomes.

### 2.1. Laboratory and Microbiological Procedure

Plasma CMV-DNA load was measured using CMV quantitative nucleic acid test on Cobas^®^ 6800 Systems (Roche Diagnostics^®^), with a lower limit of quantification of 34.5 UI/mL. According to the manufacturer’s instructions, equivalence of UI/mL to copies/mL was 0.91 UI/mL to 1 copy/mL. Galactomannan (GM) qualitative detection was performed by sandwich chemiluminescent immunoassay (CLIA) Aspergillus Galactomannan Ag VIRCLIA@ Monotest (Vircell, Spain). According to the manufacturer’s instructions, a result equal to or greater than 0.20 was considered positive in both serum and bronchoalveolar lavage. Respiratory samples for fungi cultures were grown in Sabouraud gentamicin chloramphenicol agar and antifungal sensitivity tests were performed by broth microdilution at a reference center (Carlos III National Institute). In some samples, direct visualization by KOH stain was performed. All tests were performed according to the manufacturer´s instructions. No CAPA or CMV screening protocol was implemented during the study period, and the tests were ordered at the discretion of the attending physician.

### 2.2. Definitions

CAPA was defined according to the 2020 ECMM/ISHAM criteria [18]. Cases were classified either as possible, probable, or proven CAPA. CMV replication (CMV infection) was defined as a detectable CMV-DNA on a plasma CMV PCR. CMV disease was considered proven when CMV inclusions were observed in tissue samples. CMV end-organ disease was considered probable if plasma CMV-DNA was detectable, and there was evidence of end-organ involvement according to current criteria [19]. Hematological cytopenias and hepatitis with no other alternative explanation were considered as probable CMV end-organ diseases. ARDS was defined according to Berlin’s criteria [20].

### 2.3. Primary and Secondary Objectives

Our primary objective was to compare the prevalence of CMV replication between cases and controls. Secondary objectives included comparing peak CMV-DNA load, the prevalence of patients with the highest CMV-DNA load greater than 500, 1000, and 2000 UI/mL, the prevalence of patients with probable CMV disease, and rates of attributable mortality to CMV replication in patients with and without CAPA.

### 2.4. Data Analysis

Data were presented as the median and interquartile range (IQR) for quantitative variables and as a percentage and absolute value for qualitative variables. Inferential statistical analysis was performed by means of chi-squared and Fisher exact tests (when necessary), for qualitative variables, and Mann–Whitney’s U for quantitative ones. A multivariate logistic regression model of factors associated with CAPA was performed, including as covariates CMV replication and other clinical and statistically significant variables associated with CAPA in the univariate analysis. Odds ratios (ORs) with 95% confidence intervals (CIs) were provided. Bilateral *p* values below 0.05 were considered statistically significant. All statistical analyses were performed using SPSS version 25 software (SPSS Inc., IBM, Chicago, IL, USA).

## 3. Results

Between March 2020 and August 2021, 28 CAPA patients were identified in our center, including 22 ICU admitted patients, representing 6.04% (22/364) of the patients admitted to ICU with severe ARDS due to COVID-19. After excluding patients with no concomitant plasma CMV PCR available, 24 CAPA patients were included as cases. Of these, 20 were at the ICU ward, 1 was on non-invasive mechanical ventilation at a respiratory care unit, and 3 were severe ARDS COVID-19 cases receiving the highest oxygen supplementation at conventional hospitalization. Accordingly, 48 matched controls were selected.

### 3.1. Characteristics of Patients with CAPA

Characteristics of the 24 CAPA patients included in the study are shown in Table 1. Notably, 45.8% (*n* = 11) had a previous chronic respiratory disease, and as much as 41.7% (*n* = 10) were immunocompromised prior to the COVID-19 diagnosis.

The median time from hospital admission to CAPA diagnosis was 22 days (IQR 13–47), and the median time from ICU admission to CAPA was 14 days (IQR 7–42).

Overall, 14 patients (58.3%) were classified as probable CAPA and 10 (41.7%) as possible CAPA. There were no cases of proven CAPA. Evidence of tracheobronchitis was noted in 4 out of 14 patients with available bronchoscopy (28.6%). In five cases (20.8%), serum galactomannan was positive, and two (8.3%) patients had extra-pulmonary involvement.

*Aspergillus fumigatus complex* was the most frequent species isolated (66.7%, *n* = 16), followed by Aspergillus niger complex (12.5%, *n* = 3). There were isolated cases of *Aspergillus terreus* and *Aspergillus flavus*. In three cases, there was no positive culture. In 20 cases, antifungigram was available: Only one case was resistant to amphotericin B resistance (*A. terreus*, intrinsic resistance), while no cases of resistance to voriconazole or isavuconazole were detected. Appendix A shows patient-level microbiological data and available antifungigrams. CAPA patients had an in-hospital mortality rate of 62.5% (*n* = 15), whereas controls had an in-hospital mortality of 18.8% (*p* < 0.001)

### 3.2. Association of CAPA with CMV

Table 1 summarizes baseline characteristics, COVID-19 complications and management, and outcomes, including all CMV-related data in CAPA patients and controls.

CMV replication was detected more frequently in CAPA patients than in controls (75.0% (18) vs. 35.4% (17), *p* = 0.002). Additionally, among patients with CMV replication, peak CMV-DNA load was higher in CAPA patients (median 2550 UI/mL (1092–5830) vs. 641 UI/mL (291–1805), *p* < 0.001) (Figure 1). Furthermore, probable CMV end-organ disease was more prevalent in CAPA (20.8% (5) vs. 4.2% (2), *p* = 0.037).

In the multivariate logistic regression analysis (Table 2), CMV replication was strongly associated with CAPA (unadjusted OR 5.47, 95% CI 1.82–16.38, *p* = 0.002). This persisted even after adjusting for other factors associated with CAPA in the univariate analysis (adjusted OR 7.31, 95% CI 1.41–37.83, *p* = 0.018).

### 3.3. CMV Replication among CAPA Patients

Overall, 18 (18/24, 75%) CAPA patients presented CMV replication. Table 3 summarizes the characteristics of CAPA patients with and without evidence of CMV replication. CMV replication among CAPA patients was associated with the presence of a confirmed bacterial coinfection (83.3% vs. 16.7%, *p* = 0.007).

There were no differences across CAPA categories in the prevalence of CMV reactivation (possible CAPA 71.4% (10/14) versus probable CAPA 80% (8/10), *p* = 1.000)) or peak CMV-DNA load (probable CAPA median 2540 (IQR 470–3532) vs. possible CAPA 2550 (1380–7500), *p* = 360), (Figure 2). Probable CMV end-organ disease was more frequent among probable cases (40% (4/10) vs. 7.1% (1/14)) but without reaching statistical significance (*p* = 0.075).

Regarding the timing of CMV replication, only 11/18 patients with CAPA and CMV replication had an available plasma CMV PCR determination prior to CAPA diagnosis. In 9/11 cases with available CMV PCR, it was diagnosed before CAPA. In the nine remaining cases, CMV replication was only observed after CAPA, but in seven of them who lacked a CMV determination prior to CAPA diagnosis, it is not possible to exclude an earlier reactivation. Among the nine cases that presented CMV replication prior to CAPA, the median time from CMV replication to CAPA diagnosis was 18 days (IQR 1–47).

In-hospital mortality was comparable among CAPA patients with and without CMV replication (62.5% vs. 61.1%, *p* = 1.000).

## 4. Discussion

Our results suggest that in critically ill COVID-19 patients, CMV replication is significantly more frequent among those developing CAPA than in controls without CAPA. Additionally, higher peak CMV viral load and more frequent end-organ involvement by CMV were observed in CAPA patients.

CMV infection and disease are well known to be associated with invasive aspergillosis in immunocompromised hosts, specifically solid organ transplantation recipients [9,13,21] and stem cell transplantation recipients [10,11,13]. Less is known about the association between CMV and aspergillosis outside transplantation. Nevertheless, one recent study showed the association between CMV and aspergillosis occurs as well in critically ill patients [12]. However, to the best of our knowledge, this association has not been confirmed to date in any published study in patients with COVID-19 and invasive pulmonary aspergillosis. In the present study, 75% of CAPA patients had CMV DNAemia, a percentage far beyond that observed in critically ill COVID-19 controls. The prevalence of CMV replication in our CAPA patients was also higher than what was found by other authors in critically ill COVID-19 patients without CAPA [15].

The pathogenesis of the association of CMV and invasive aspergillosis is not fully understood. On the one hand, several authors have pointed to the immunomodulatory effect of CMV infection, especially the dysregulation in T-lymphocyte function [21] and antigen presentation [13]. It has been proposed that the immunomodulatory effects of CMV infection convey a greater risk of secondary infection, including fungal superinfections. Even low levels of viruses that may not complete a full replicative cycle are able to display CMV antigens on cells, thus becoming targets for immune-pathological responses [22]. On the other hand, CMV replication and invasive aspergillosis share several factors that may favor both of them, such as immunocompromised status, and specifically chronic corticoid use [23]. Polymorphisms in the innate immune system Toll-like receptors could contribute to the development of both infections [13].

In critically ill COVID-19 patients, several factors may favor aspergillosis development, including inflammatory pulmonary damage [4] and immunosuppressive drugs used to treat the disease, especially anti-IL-6 drugs [6]. In addition, several authors have emphasized the importance of immune response dysregulation in those patients [24], with a focus on T-cell dysfunction and paralysis [25], which has been related to CAPA pathogenesis [7].

Consequently, although a cause–effect relationship remains to be demonstrated, T-cell dysfunction originated by both COVID-19 and CMV could have a synergistic effect in the susceptibility and development of invasive aspergillosis in critically ill COVID-19 patients, explaining the high proportion of CAPA patients with CMV replication and end-organ disease found in our study.

Interestingly, in our study, in patients with available serum CMV RT–PCR prior to CAPA diagnosis, CMV replication usually preceded CAPA (9 out of 11 cases], in line with what is found in other immunocompromised patients [10,14].

The pathogenic role of CMV viremia is unclear, and in many cases, it has been considered an innocent bystander, or an indicator of immunocompromised condition [13]. Only a small proportion of patients with CMV viremia develop end-organ diseases, owing to the effect of immune responses at each organ in preventing it. If the immune response is insufficient, CMV may rise to high levels and cause end-organ disease [22]. We hypothesize that CMV DNAemia is a marker of immunocompromised in COVID-19 patients and may warn of the risk of developing CAPA. In this scenario, surveillance of CMV DNAemia in COVID-19 patients could inform of patients who would benefit from a tighter control and potentially targeted antifungal prophylaxis. Another issue that deserves evaluation is the effect of CMV treatment in the subsequent development of CAPA.

Ours is a single-center, retrospective study and has the inherent limitations of this design. Another limitation is that in our institution, there was not a CAPA screening protocol, and consequently, a respiratory fungal culture was not available for every control patient, so it is not possible to exclude that a small proportion of the controls actually had CAPA. However, the prevalence of CAPA in our institution is similar to the one found in similar settings [26] as well as in recently systematic reviews [2,3], suggesting that the majority of CAPA cases were identified. Secondly, we could not analyze the timing from CMV replication to CAPA in all of the cases included in the study, since not all patients had an available CMV RT–PCR prior to CAPA diagnosis. Nevertheless, the afore-mentioned possible pathogenic relationship between these entities and our data suggest that CMV replication most often preceded CAPA diagnoses. Finally, similar to other studies on CAPA, the differentiation between *Aspergillus spp* colonization and invasive disease in critically COVID-19 patients is not obvious, given that histologic samples are rarely available and clinical–radiological features are often overlapping and non-specific. However, we made efforts to mitigate this limitation by systematically applying the 2020 ECMM/ISHAM consensus criteria for CAPA diagnosis and classification.

## 5. Conclusions

Among critically ill COVID-19 patients, CMV infection was associated with CAPA and could potentially be considered a harbinger of CAPA. Being aware of this association is important to prompt diagnosis of CAPA in patients with CMV replication. Further studies are needed to confirm this association and determine the role of CMV treatment.

## Figures and Tables

**Figure 1 jof-08-00161-f001:**
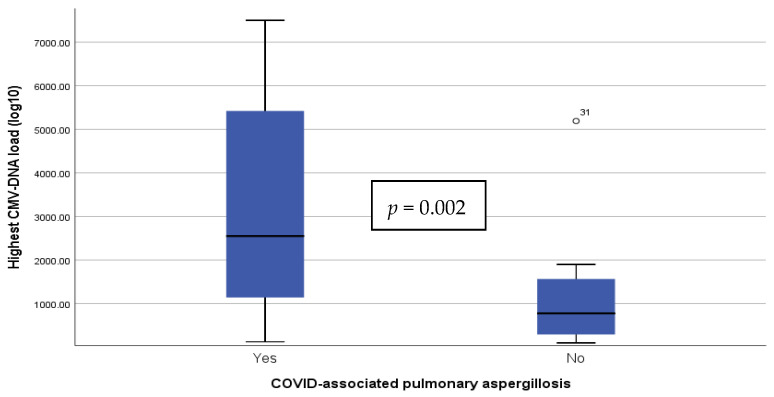
Peak CMV-DNA viral load among patients with CMV replication according to the diagnosis of pulmonary aspergillosis. Viral load is expressed in UI/mL (0.91 UI/mL equals 1 copy/mL). and represented in a linear scale. CMV: cytomegalovirus.

**Figure 2 jof-08-00161-f002:**
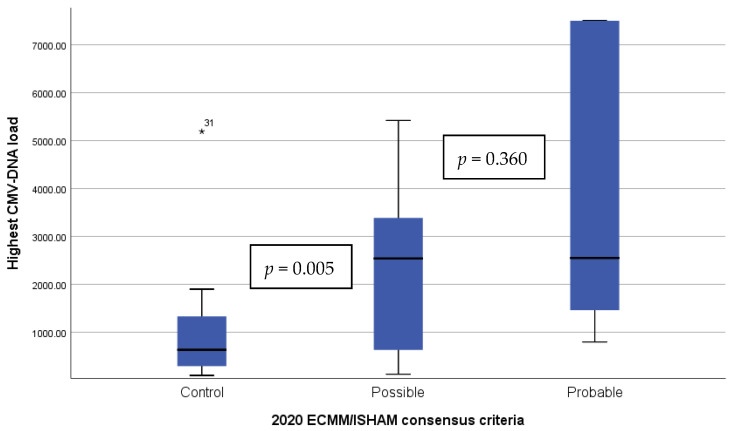
CMV-DNA viral load in patients with CMV replication according to 2020 ECMM/ISHAM consensus criteria classification of COVID-associated pulmonary aspergillosis and controls. Viral load is expressed in UI/mL (0.91 UI/mL equals 1 copy/mL) and represented in in a linear scale. CMV: cytomegalovirus. * represents an individual patient with a unusual high CMV-DNA viral load within controls.

**Table 1 jof-08-00161-t001:** Factors associated with CAPA versus controls.

Variable	CAPA (*n* = 24)	Control (*n* = 48)	*p*	Missing
COMORBIDITY				
Age (years)	68 (65–72)	61 (54–70)	0.014	0
Sex (female)	20.8% (5)	27.1% (13)	0.774	0
Active smoking	22.2% (4/18)	2.4% (1/41)	0.026	13
Alcohol abuse	25.0% (3/12)	3.8% (1/26)	0.084	34
Arterial hypertension	62.5% (15)	45.8% (22)	0.217	0
Diabetes mellitus	37.5% (9)	12.5% (6)	0.028	0
Chronic respiratory disease	45.8% (11)	31.3% (15)	0.299	0
*COPD*	29.2% (7)	10.4% (5)	0.050	0
*Asthma*	4.2% (1)	10.4% (5)	0.656	0
*Other*	20.8% (5)	10.4% (5)	0.285	0
Chronic cardiac failure	20.8% (5)	6.3% (3)	0.107	0
Ischemic heart disease	20.8% (5)	2.1% (1)	0.014	0
Chronic renal failure	29.2% (7)	4.2% (2)	0.005	0
Liver cirrhosis	4.2% (1)	0	0.333	0
Solid organ malignancy	8.3% (2)	6.3% (3)	1.000	0
PRIOR IMMUNOCOMPROMISED STATUS				
Any IC	41.7% (10)	18.8% (9)	0.049	0
Hematological malignancy	16.7% (4)	4.2% (2)	0.091	0
Solid organ transplantation	16.7% (4)	2.1% (1)	0.039	0
HSCT	0	0	-	0
Autoimmune disease	12.5% (3)	10.4% (4)	1.000	0
Previous chronic corticoid	25.0% (6)	4.2% (2)	0.014	0
Other previous IS treatments	25.0% (6)	10.4% (5)	0.163	0
COVID-19 PRESENTATION AND MANAGEMENT PRIOR TO CAPA DIAGNOSIS				
Neutropenia	16.7% (4)	2.1% (1)	0.039	0
Confirmed bacterial coinfection	66.7% (16)	64.6% (31)	1.000	0
Viral coinfection **	8.3% (2)	2.1% (1)	0.546	0
Renal substitutive therapy	37.5% (9)	16.7% (8)	0.076	0
Vasopressor drug therapy	41.7% (10)	56.3% (27)	0.319	0
APACHE II	12 (9–19)	10 (8–13)	0.032	2
Any corticoid treatment	100%	97.9% (46)	0.546	0
Corticoid pulses	54.2% (13)	35.4% (17)	0.204	0
Tocilizumab	95.8% (23)	64.6% (31)	0.004	0
*1 dose*	45.0% (9/20)	90.3% (28/31)	0.001	3
*2 or more doses*	55.0% (11/20)	9.7% (3/31)		
Anakinra	12.5% (3)	10.4% (5)	1.000	0
Remdesivir	16.7% (4)	6.3% (3)	0.212	0
Antibiotics	95.8% (23)	100% (48)	0.333	0
CYTOMEGALOVIRUS (CMV) REACTIVATION				
CMV reactivation	75.0% (18)	35.4% (17)	0.002	0
*More than 500 UI/mL*	66.7% (16)	23.4% (11)	0.001	0
*More than 1000 UI/mL*	58.3% (14)	12.8% (6)	<0.001	0
*More than 2000 UI/mL*	45.8% (11)	4.3% (2)	<0.001	0
Peak CMV-DNA load *	2550 (1092–5830)	641 (291–1805)	0.002	37 *
CMV end-organ disease	20.8% (5)	4.2% (2)	0.037	0
MORTALITY AND OUTCOMES				
In-hospital mortality	62.5% (15)	18.8% (9)	<0.001	0
ICU length of stay	61 (37–89)	33 (18–62)	0.024	27
Hospital length of stay	74 (56–97)	50 (27–76)	0.077	25

*: Median CMV viral loads are calculated considering only those patients with detectable serum CMV DNA. Viral load is measured in UI/mL (0.91 UI/mL equals 1 copy/mL). ** Viral coinfection was one case of serum herpes-simplex-1 replication and one case of serum Epstein–Barr virus replication. CAPA: COVID-associated pulmonary aspergillosis; COPD: chronic obstructive pulmonary disease; HSCT: hematopoietic stem cell transplantation; IC: immunocompromised IS: immunosuppressive.

**Table 2 jof-08-00161-t002:** Multivariate analysis of factors associated with COVID-associated pulmonary aspergillosis.

Variable	Adjusted OR	95% CI	*p*
CMV replication	7.31	1.41–37.83	0.018
Age (per year)	1.04	0.96–1.11	0.300
Diabetes mellitus	5.91	0.9835.62	0.053
Chronic renal failure	6.36	1.09–118.4	0.042
Any immunosuppressive condition	0.81	0.14–4.51	0.809
Pulse doses of corticoid	1.30	0.28–5.99	0.733
Tocilizumab	14.30	1.21–192	0.035
APACHE II (per point)	1.12	0.99–1.27	0.082

Unadjusted OR for CMV replication: 5.47, 95% CI 1.82–16.38, *p* = 0.002. Multivariate analysis was conducted by means of a logistic regression model with conditional backward variable exclusion. CMV: cytomegalovirus. Active smoking was not introduced due to the high number of missing data.

**Table 3 jof-08-00161-t003:** CMV replication-associated factors among CAPA patients with serum CMV-DNA available.

Variable		Total (*n* = 24)	CMV Replication (*n* = 18)	No CMV Replication (*n* = 6)	*p*
COMORBIDITY					
Age (years)		68 (65–72)	69 (65–73)	68 (60–71)	0.494
Sex (female)		20.8% (5)	22.2% (4)	16.7% (1)	1.000
Active smoking		16.7% (4)	11.1% (2)	33.3% (2)	0.217
Chronic respiratory disease		45.8% (11)	38.9% (7)	66.7% (4)	0.357
Chronic renal failure		29.2% (7)	22.2% (4)	50.0% (3)	0.307
Any IC condition		41.7% (10)	44.4% (8)	33.3% (2)	1.000
Hematological malignancy		16.7% (4)	16.7% (3)	16.7% (19	1.000
Solid organ transplant		16.7% (4)	16.7% (3)	16.7% (1)	1.000
Previous chronic corticoid		25.0% (6)	27.8% (5)	16.7% (1)	1.000
Other previous IS treatments		25.0% (6)	22.2% (4)	33.3% (2)	1.000
COVID-19 PRESENTATION AND MANAGEMENT PRIOR TO CAPA DIAGNOSIS					
Confirmed Bacterial coinfection		66.7% (16)	83.3% (15)	16.7% (1)	0.007
Antibiotic treatment		95.8% (23)	100% (18)	83.3% (5)	0.250
Renal replacement therapy		37.5% (9)	33.3% (6)	50.0% (3)	0.635
Vasopressor drug		41.7% (10)	38.9% (7)	50.0% (3)	1.000
APACHE II		12 (9–19)	13 (9–19)	11 (10–18)	0.923
Corticoid pulses		54.2% (13)	55.6% (10)	50.0% (3)	1.000
Tocilizumab 2 doses		45.8% (11)	44.4% (8)	50.0% (3)	1.000
Blood transfusion		60.9% (14)	58.8% (10)	66.7% (4)	1.000
ASPERGILLOSIS RADIOLOGY AND CLINICAL PRESENTATION					
Days from admission		22 (13–47)	22 (12–56)	21 (14–25)	0.626
Days from ICU admission		14 (7–42)	19 (6–42)	8 (7–16)	0.349
Tracheobronchitis		28.6% (4/14)	18.2% (2/11)	66.7% (2/3)	0.175
Solitary nodule		12.5% (3)	16.7% (3)	0	0.546
Multiple nodules		20.8% (5)	16.7% (3)	33.3% (2)	0.568
Cavitary nodule (s)		25.0% (6)	33.3% (6)	0	0.277
Alveolar infiltrate		70.8% (17)	72.2% (13)	66.7% (4)	1.000
ASPERGILLOSIS MICROBIOLOGY					
*A. fumigatus complex*		66.7% (16)	61.1% (11)	83.3% (5)	0.878
*A. niger complex*		12.5% (3)	11.1% (2)	16.7% (1)	
Other species		8.4% (2)	11.2% (2)	0	
No culture growth		12.5% (3)	16.7% (3)	0	
ASPERGILLOSIS CLASSIFICATION					
2020 ECMM criteria	*Probable*	41.7% (10)	44.4% (8)	33.3% (2)	1.000
	*Possible*	58.3% (14)	55.6% (10)	66.7% (4)	
OUTCOMES					
In-hospital mortality		62.5% (15)	61.1% (11)	66.7% (4)	1.000
ICU length of stay		61 (37–89)	62 (33–98)	Not applicable	-
Hospital length of stay		74 (56–97)	82 (63–117)	Not applicable	-

Qualitative variables are expressed as percentage (absolute number). Quantitative variables are expressed as median (interquartile range). CMV: cytomegalovirus; CAPA: COVID-associated pulmonary aspergillosis; IC: immunocompromised; ICU: intensive care unit.

## Data Availability

The data presented in this study are available on request from the corresponding author.

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
