# Peer review of "Association of COVID-19-Associated Pulmonary Aspergillosis with Cytomegalovirus Replication: A Case–Control Study"

_jof, 2022, doi:10.3390/jof8020161_

Round 1

Reviewer 1 Report

The authors report a case control study of CAPA and the association of CMV.

The paper is presented well.

Major comments:

1.The paper describes the association of invasive aspergillosis among critically ill subjects with CMV disease; the only, 'novel' factor is that COVID-19 is the critical illness in the current study. hence, the paper reaffirms the prevalent knowledge that CMV reactivation in critically ill individuals predisposes to infections and the same holds true for COVID-19. If the authors had a severity matched non COVID critically ill subjects, the contribution of COVID-19 could be assessed.

2.While the authors currently include factors significant on univariate analysis for the regression analysis. "any previous IC" and "previous corticosteroids" are overlapping and their interaction might have resulted in the current results. Rather the authors should retain only "any previous IC" since that would include SOT, HM and corticosteroids, all of which can result in CMV .

In addition, the authors may include "pulse doses of corticosteroids" in the model, (even though the difference is not statistically significant, it is clinically relevant). It is possible that diabetes, pre exisiting immunosuppression, tocilizumab and pulse doses of corticosteroids predispose to both CMV as well as Aspergillosis.

Minor comments:

Table 1: "viral coinfection" - what are they?

Lines 62-63: 2017??

Author Response

We thank the reviewers for their comments, that we believe that improve the paper.  Please, see below our point-by-point response.

Reviewer 1:

The authors report a case control study of CAPA and the association of CMV. The paper is presented well.

Major comments:

1.The paper describes the association of invasive aspergillosis among critically ill subjects with CMV disease; the only, 'novel' factor is that COVID-19 is the critical illness in the current study. hence, the paper reaffirms the prevalent knowledge that CMV reactivation in critically ill individuals predisposes to infections and the same holds true for COVID-19. If the authors had a severity matched non COVID critically ill subjects, the contribution of COVID-19 could be assessed.

The reviewer brings here an interesting topic. The CMV reactivation in critically ill patients is not new, however, its  influence in outcome is a matter of controversy.( Li X, Huang Y, Xu Z, Zhang R, Liu X, Li Y, Mao P. Cytomegalovirus infection and outcome in immunocompetent patients in the intensive care unit: a systematic review and meta-analysis. BMC Infect Dis. 2018 Jun 28;18(1):289 .

In particular, the association of CMV  infection and invasive aspergillosis in critically-ill patients, has only been described very recently by Kuo and colleagues (Kuo CW, Wang SY, Tsai HP, Su PL, Cia CT, Lai CH, Chen CW, Shieh CC, Lin SH. Invasive pulmonary aspergillosis is associated with cytomegalovirus viremia in critically ill patients - A retrospective cohort study. J Microbiol Immunol Infect. 2021 Mar 31:S1684-1182(21)00057-8), To the best of our knowledge, this is the only paper that describes this association in the critically ill setting so far.

On the contrary, the association between CMV and aspergillosis has been firmly demonstrated in highly immunocompromised hosts such as stem cell and solid organ transplantation recipients. The inter-relationship between these infections has not been comprehensively explored, and there still remain a number of research questions (Yong MK, Slavin MA, Kontoyiannis DP. Invasive fungal disease and cytomegalovirus infection: is there an association? Curr Opin Infect Dis. 2018 Dec;31(6):481-489). Among them, the possible role of viral co-infection (e. g. CMV and SARS-CoV-2) in increasing the risk of aspergillosis. We hypothesized that CMV infection in COVID-19 patients would add to the risk for CAPA, even in otherwise healthy individuals that lack the classical risk factors for invasive aspergillosis.

Based on these data, we decidedly believe that our work adds valuable information to current literature, being the first to describe the association of CMV and IA in COVID-19 patients. 

We agree with the reviewer that including non-COVID critically ill subjects would allow to assess the contribution of COVID-19 to invasive pulmonary aspergillosis development, but this was not the purpose of the present study. On the contrary, our aim is to evaluate the contribution of CMV infection to the development of CAPA among COVID-19 patients. Thus, the denominator in the present study are the critically ill COVID-19 patients, and not the critically ill patients with CMV reactivation. We intend to study the role of CMV-SARS-CoV2 coinfection in the development of invasive aspergillosis. This has been clarified in the manuscript.

2.While the authors currently include factors significant on univariate analysis for the regression analysis. "any previous IC" and "previous corticosteroids" are overlapping and their interaction might have resulted in the current results. Rather the authors should retain only "any previous IC" since that would include SOT, HM and corticosteroids, all of which can result in CMV .

In addition, the authors may include "pulse doses of corticosteroids" in the model, (even though the difference is not statistically significant, it is clinically relevant). It is possible that diabetes, pre exisiting immunosuppression, tocilizumab and pulse doses of corticosteroids predispose to both CMV as well as Aspergillosis.

Response: Thank you for this valuable suggestion. As proposed, we have retained only “any previous IC”. Additionally, we have included the variable “pulse doses of corticosteroids”. We have updated table 2 with the new variables and data. As shown in the table, the independent association of CMV replication and CAPA persists after adjusting the multivariate regression model with the new variables.

Minor comments:

Table 1: "viral coinfection" - what are they?

Response: There was one case of HSV-1 replication and one case of EBV replication, both in serum. We have added this information in the footnote of table 1.

Lines 62-63: 2017??

Response: The Hospital Universitario Puerta de Hierro-Invasive Fungal Infection (HUPH-IFI) cohort is a prospective ongoing registry of all invasive fungal infections diagnosed at our institution. It includes not only CAPA cases, but all kinds of invasive fungal infections. It is maintained by means of constant communication between microbiology department and clinical infectious diseases staff. We have identified CAPA cases from this cohort. This registry began in 2017, prior to the COVID-19 pandemic.

Reviewer 2 Report

The manuscript #jof-1555325, entitled “Association of COVID-19-associated pulmonary aspergillosis with cytomegalovirus replication: a case-control study” by Calderón-Parra et al. presents a comprehensive article on, in line with the title, the association between cytomegalovirus (CMV) replication and COVID-19-associated pulmonary aspergillosis (CAPA) occurrence. The article is very well presented - I appreciate logical design of the study, logical design of the manuscript itself, as well as aesthetics such as almost none editorial mistakes. Also, I appreciate the fact that all necessary informations are included in the introduction and discussion sections. Both sections have anticipated most of my questions, which originated during exploring the manuscript. 

Major issue:

The authors stated that "In 20 cases the antifungigram was available: only 1 case was resistant to amphotericin B resistance (A. terreus, intrinsic resistance), while no cases of resistance to voriconazole or isavuconazole were detected." (L137-139). Please provide the antifungal data for each isolate either as a Table or graphs in the main text or as a Supplementary materials.

Minor issues:

Please pay attention to correct all Latin names (e.g. Aspergillus) into italics.

L145: "Table 1. Table 1:" please correct.

Author Response

We thank the reviewers for their comments, that we believe that improve the paper.  Please, see below our point-by-point response.

Reviewer 2

The manuscript #jof-1555325, entitled “Association of COVID-19-associated pulmonary aspergillosis with cytomegalovirus replication: a case-control study” by Calderón-Parra et al. presents a comprehensive article on, in line with the title, the association between cytomegalovirus (CMV) replication and COVID-19-associated pulmonary aspergillosis (CAPA) occurrence. The article is very well presented - I appreciate logical design of the study, logical design of the manuscript itself, as well as aesthetics such as almost none editorial mistakes. Also, I appreciate the fact that all necessary informations are included in the introduction and discussion sections. Both sections have anticipated most of my questions, which originated during exploring the manuscript. 

Major issue:

The authors stated that "In 20 cases the antifungigram was available: only 1 case was resistant to amphotericin B resistance (A. terreus, intrinsic resistance), while no cases of resistance to voriconazole or isavuconazole were detected." (L137-139). Please provide the antifungal data for each isolate either as a Table or graphs in the main text or as a Supplementary materials.

Response: Thank you for the appreciation and the valuable suggestion. We have added microbiological and antifungal data as a table in supplementary material (table S1).

Minor issues:

Please pay attention to correct all Latin names (e.g. Aspergillus) into italics.

Response: Thank you for the advice. We have checked Latin names and changed into italics when indicated.

L145: "Table 1. Table 1:" please correct.

  Response: Corrected.

Round 2

Reviewer 1 Report

My questions have been addressed

Minor error as below, needs rectification

"adjusted OR 8.28, 95% CI 1.90-36.13, p=0.005" in text line 166, needs to be corrected as per the new Table 2

Author Response

My questions have been addressed

We thank the reviewers for their consideration of our paper and accepting previous responses and corrections, which have improved our manuscript.

Minor error as below, needs rectification

"adjusted OR 8.28, 95% CI 1.90-36.13, p=0.005" in text line 166, needs to be corrected as per the new Table 2

We thanks the reviewer for noticing the error. We have corrected the text as per the new Table 2 as indicated.

Reviewer 2 Report

I accept all the corrections.

Author Response

We thank the reviewers for their consideration of our paper and accepting previous responses and corrections, which have improved our manuscript.